Wind Energ. Sci. Discuss., 2020, 1-20, 10.5194/wes-2020-15, 2020.

# Revealing system variability of offshore service operations through systemic hazard analysis

Romanas Puisa[a], Victor Bolbot[a], Andrew Newman[b], Dracos Vassalos[a]

[a]Maritime Safety Research Centre, University of Strathclyde, UK
[b]Global Marine Group, UK

*Correspondence to:* Romanas Puisa (r.puisa@outlook.com)

**Abstract.** As windfarms are moving further offshore, logistical concepts increasingly include service operation vessels (SOV) as the prime means of service delivery. However, given the complexity of SOV operations in hostile environments, their safety management is challenging. The objective of this paper is twofold. First, we perform a systemic hazard analysis by the STPA method for three phases of SOV operation: when transiting and manoeuvring within a windfarm, interfacing with turbines, and launching or recovering daughter crafts. This gives us sets of scenarios containing potentially hazardous interactions between various system components. Such scenarios reflect the complexity and potential for necessary and unwanted variability in the system. Second, we use these results to compare the three operational phases in terms of a proposed systemic indicator—the system variability. The comparison shows that all three phases of SOV operation have rather comparable levels of variability. However, the interface between the SOV and turbine via the gangway system and the manoeuvring between turbines seem to show a higher potential for uncontrolled variability. We have broadly discussed what resources and capabilities should be added to improve the control. The study has also shown how results of a systemic hazard can be linked to systemic indicators or measures of system safety, namely to the system variability.

## 1   Introduction

Offshore wind is becoming a major source of renewable energy in many countries (GWEC, 2019). As wind farms are moving further offshore, significant innovations in the infrastructure and services are required to maintain the judicious trend. One of such innovations is the specialised service vessels, or service operation vessels (SOVs), which are offering new logistical concepts for servicing windfarms further offshore. They enable an extended stay of technicians (typically for two weeks) in the vicinity of a windfarm, thereby replacing the logistical concept of transferring technician from shore by crew transfer vessels (CTVs). The latter becomes unreasonable due to prolonged sailing times and increased risk of seasickness.

SOVs are akin to offshore supply vessels and are typically around 80 meters in length, can endure severe environmental conditions and offer a wide array of services. They are highly automated ships (e.g., position and course can be kept automatically by the Dynamic Positioning (DP) system), hosting dozens of technicians, support (daughter) crafts, and heavy

Wind Energ. Sci. Discuss., 2020, 1-20, 10.5194/wes-2020-15, 2020.

equipment. Daughter crafts (DCs) are medium size boats, typically under 20 meters, which are carried by the SOV and used
to transport lighter equipment to turbines in moderate environmental conditions (< 1.8m significant wave height). DCs are
loaded with technicians and launched from a SOV deck by some davit system, typically 3-5 times per day, and then recovered
(lift-up) from the water periodically. SOVs would also have a sophisticated system for transferring technicians and equipment
to and from a turbine. It is normally a motion-compensated (3 or 6 DoF) gangway system, which allows for relatively safer
(based on experience so far) and time-efficient (within some 5 minutes) transfer.
The multifaceted nature of SOV operations complicates the management of their safety. The overall safety management of
SOV operations is an amalgamation of individual safety procedures for the SOV, davit, DC, gangway, drone and other systems
(Section 2). These safety systems are developed in isolation from a wider operational context and, when integrated, can lead
to confusion, surprises and undue pressure on operators (Ahsan et al., 2019). In such conditions, accidents can be caused by
well-known but inadequately managed scenarios (e.g., loss of power or control), as well as by yet unknown scenarios created
by new technology or new ways of operation. In 2018, the offshore supply vessel Vos Stone temporally lost control of thrusters,
drifted and struck a wind turbine (BSU, 2019). Amongst the causes, the officers on the bridge did not manage to seamlessly
switch between modes of thruster control (from DP to other mode) because they were confused about them. Inadequately
controlled transitions between modes of operation, particularly between normal (frequently used) and abnormal (rarely used,
e.g. emergency) modes, is a classic scenario for accidents (Sarter et al., 1997;Leveson, 2011a, p. 289). Another incident
happened in 2013 when the diving support vessel Bibby Topaz drifted off the position (maintained by the DP system) while
two divers where exploring the seabed (IMCA, 2013). Amongst the causes, the vessel had had a dormant (unidentified)
hazard—a design error—that did not allow to adequately respond to safety critical faults that preceded the incident.
As the cost-efficiency of marine operations is being increased by more automation and autonomy (Twomey, 2017), systems
become more tightly coupled, nonlinear and more difficult to understand, design, analyse and operate (Perrow, 1984). Such
complex systems tend to create "interactions in an unexpected sequence" (Perrow, 1984, p. 78), and some of these interactions
can be hazardous. These interactions, and their consequences, is difficult to envisage during design of individual system
components or sub-assemblies, because they manifest themselves at the system level and under specific circumstances or
scenarios. System properties and events such as safety, or absence thereof, are emergent (Checkland, 1981;Meadows,
2008;Leveson, 2011b), or as Rasmussen put it "a system is more than the sum of its elements" (Rasmussen, 1997). We,
therefore, cannot predict when untoward events (near-misses, incidents, and accidents) will occur, as we used to do with
electromechanical systems with well-defined probability distributions of failures.
This means that focusing on individual hazardous scenarios or hazards is oversimplification and is unhelpful. Safety is a system
property, which cannot be inferred from properties of individual components or scenarios (Leveson, 2011b). By making this
assumption we follow the systemic view (systems thinking) on safety as highlighted above and acknowledge the danger of the

Wind Energ. Sci. Discuss., 2020, 1-20, 10.5194/wes-2020-15, 2020.

reification fallacy, i.e. "the tendency to convert a complex process or abstract concept into a single entity or thing in itself"
(Gould and Gold, 1996;Hollnagel and Woods, 2006). One, therefore, should regard hazards and related events are symptoms
of wider, structural issues within safety control. They are symptoms that the safety control structure in place is inadequately
designed, because it should not lead to hazards otherwise (Leveson, 2011a, p. 100).
Consequently, instead of trying to infer safety or some risk level from individual hazards and scenarios, a systemic measure
should be used. As discussed in Section 4.1, this measure should reflect the variability within the system where technical and
human components interact. This variability can then combine with other sources of variability such as latent conditions (e.g.,
negligent safety culture, inadequate feedback on system performance) and impaired or missing safety barriers to lead to unsafe
system states or hazards (Hollnagel, 2016). By knowing where and when the system variability is highest, changes to the
design and operation can be made to control it better and, subsequently, prevent incidents and accidents.
With the above in mind, the objective of this paper is twofold. First, we perform a systemic hazard analysis for three phases
of SOV operation: when transiting and manoeuvring within a windfarm, interfacing with turbines, and launching or recovering
daughter crafts. Second, we use the analysis results to estimate and then compare the system variability within the system
during three operational phases. For the systemic hazard analysis, we use the Systems Theoretic Process Analysis (STPA)
(Leveson, 2011a;Leveson and Thomas, 2018). This is method is based on a systemic accident model and allows to explore
hazardous scenarios caused by flawed interactions between system components and, to a lesser extent, by component failures.
The paper is organised as follows. Section 2 explores related work, Section 3 explains the basics behind safety management
currently in practice, Section 4 introduces to the concept of system variability and describes its adopted indicator, Section 5
outlines the analysis results, which are subsequently discussed in Section 6. Section 8 concludes the paper.

## 2    Related work

In this section we review academic and industrial literature on hazard, system variability and resilience analysis of servicing
windfarms and other offshore installations by SOV-like vessels.
The reviewed literature focuses on collision (ship to ship, shop to turbine), reliability issues with technology (DP, gangway,
and other systems) and human factors (Presencia and Shafiee, 2018), (Dong et al., 2017), (Rollenhagen, 1997;Sklet, 2006),
(Rokseth et al., 2017), and (SgurrEnergy, 2014). The used hazard analysis mainly followed a conventional, non-systemic,
approach where individual hazards or scenarios are considered in isolation. In most cases, statistics or probabilistic analysis is
used for decision making. The exception is Rokseth et al. who applied the STPA method to hazard analysis of offshore supply
vessels running on the DP system (Rokseth et al., 2017). None of the studies use systemic indicators or measures (e.g., of
resilience) to infer the safety level or compare operational phases or other aspects.

Wind Energ. Sci. Discuss., 2020, 1-20, 10.5194/wes-2020-15, 2020.

When it comes to indicators or measures of system variability and resilience, the general literature is abound, e.g. (Hollnagel
et al., 2007;Herrera et al., 2010). The literature specific to the maritime domain is limited but present, e.g. (Praetorius et al.,
2015;Patriarca and Bergström, 2017;de Vries, 2017). However, the authors have not come across a work which connects
results of a systemic hazard analysis, namely hazardous scenarios, with the system variability or similar systemic indicators.

## 3    Safety management practice

As any safety critical system, SOVs comply with international and national safety standards during vessel design, construction
and operation (Grace and Lee, 2017). The latter is "managed by vessel operators as part of their safety management system"
(IMCA, 2015). The key element of safety management is a risk assessment (IMCA, 2014;Bromby, 1995), i.e. the identification
of safety hazards to ships, personnel and the environment and establishment of appropriate controls. This also constitutes one
of the objectives of the International Safety Management (ISM) Code (IMO, 2018). Risk Assessment Method Statements
(RAMS) are documents that OEMs (e.g., of davit system, daughter crafts) create after they conduct individual risk assessments.
RAMS contain details on identified hazards as well as a step-by-step safe working guide that crew, contractors (technicians),
and others should follow to avoid and adequately respond to hazards. The hazards inform training, briefing notes and
operational procedures. Notably, RAMS are used interchangeably with safety procedures and manuals.
As SOV operations use diverse systems (davits, gangways, daughter crafts, drones) that interact, separate RAMS are used for
each interaction, with a bridging document to state the overall emergency protocol and document primacy (cf. Figure 1). In
other words, the overall safety management system (SMS), or safety governance, onboard of a SOV is comprised of multiple
RAMS, depending on the type of systems in interaction.

Wind Energ. Sci. Discuss., 2020, 1-20, 10.5194/wes-2020-15, 2020.

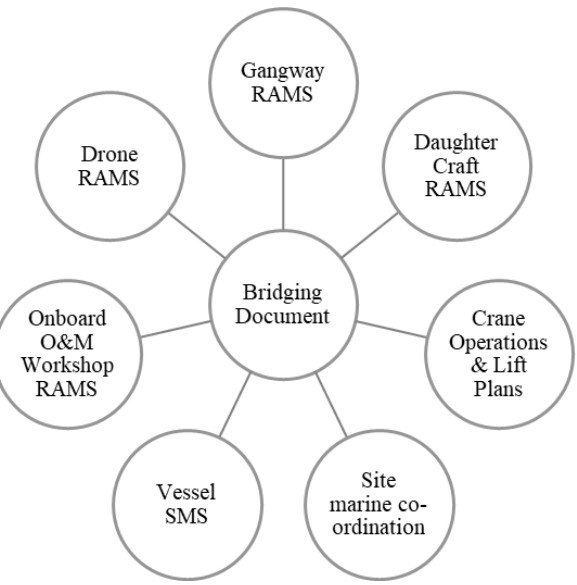


*Figure 1: Illustration of current safety governance*

For example, for a typical 14-day SOV operation in the UK, the safety governance may involve over five regulators simultaneously when alongside a turbine (Table 1). This ad-hoc or case-by-case safety management, however, happens sufficiently rare is that the developed SMS could often be timed for longer periods. This is a result of evolutionary process where a limited "bolt on" capacity was mobilised to a vessel which did not warrant a rework of the vessel safety systems.

When faced with the multitude of internal RAMS (procedures), the opportunity for confusion and hazardous surprises arises. This is because the knowledge of all individual safety procedures is often outside of what is normally expected of seafarers. Also, RAMS are developed in isolation and their amalgamation into one system can create conflicts between safety procedures or create unintended consequences. Therefore, safety management is heavily reliant on operator's general competence and familiarity with operations.

In view of these practices, a systemic, top-down approach to hazard analysis—when multiple systems (e.g., the DP and gangway systems) are engaged at the same time—is required to properly address the system-level hazards. The following section explicates why and how systemic analysis is performed.

Wind Energ. Sci. Discuss., 2020, 1-20, 10.5194/wes-2020-15, 2020.


*Table 1: Safety governance in various stages of operation*

| Stage of operation | Safety rules, regulations, RAMS |
|---|---|
| Entering the site | Marine Co-ordination rules (site specific operator rules) |
| Within exclusion zone of a turbine. | Electrical safety rules, UK MCA for port state, vessel flag state, classification society, marine co-ordination and turbine specific control centre |
| Transit from turbine to turbine | Special Purpose Ships (SPS) Code (UK MCA, class rules and flag regulations) |
| Interface with turbine | Vessel operations governed by SPS Code, crane operations by UK HSE Lifting Operations and Lifting Equipment Regulations 1998 (LOLER) regulations, workshop activities by Provision and Use of Work Equipment Regulations 1998 (PUWER), UK HSE regulations, and IMCA guidelines (IMCA, 2014) |
| Interface with daughter craft | Class rules, site specific rules, company and vessel specific guidelines |


**4    Method**
**4.1    System variability**
As argued in Section 1, a top-down, systemic approach to safety analysis should be used for complex systems. In this section
we highlight further characteristics of such systems and introduce the notion of system variability.
First, it is helpful to resume our discussion on how untoward events happen in complex, socio-technical systems. The presence
of a systematic, as opposed to random, drift to failure is characteristic for all socio-technical systems (Rasmussen,
1997;Dekker, 2016). People, and organisations, are constantly looking for trade-offs between production pressures, individual
preferences and safety expectations. Sometimes, production pressures, in particular, can outweigh the other ones, with small
deviations from earlier established norms being systematically normalised—confusing the absence of risk evidence with the
absence of risk. Hollnagel refers to this constant optimisation of performance as the ETTO principle, Efficiency-Thoroughness
Trade-Off (Hollnagel, 2017). An important element of the performance optimisation is the need to adjust to irregularities and
deviations in the expected performance of other system components and the environment. This can lead to repeated and quite
rational violations of procedures to maintain safety (Besnard and Hollnagel, 2014;Fujita, 1991;Rasmussen and Suedung, 2000).
As design errors are frequent and procedures are often underspecified in complex systems, such performance adjustments

Wind Energ. Sci. Discuss., 2020, 1-20, 10.5194/wes-2020-15, 2020.

become vital for safety and other system objectives. Hence, inability to adequately adjust to operational complexity due to
meagre resources (time, knowledge, competence, etc.) is a harbinger of untoward events (Woods and Hollnagel, 2017). By
contrast, the ability to anticipate, adjust and adapt to various irregularities is referred to as system *resilience* (Hollnagel et al.,
144    2007).

Under certain conditions the system variability can get out of control, putting safety, or performance alike, in jeopardy. This
happens when various sources of variability combine in a nonlinear fashion (i.e., when a cause is disproportional to an effect),
exploding the operational complexity and creating opportunities for hazards and, if not mitigated, for incidents and accidents.
Hollnagel defines the following sources of variability that may combine (Hollnagel, 2016, p. 170):
• Human performance variability, as explained by the ETTO principle above.
• Technological glitches, gradual and outright failures, caused by design or maintenance errors.
• Inadequate or missing safety barriers due to design or maintenance errors.
• Latent conditions such a deficient safety culture (i.e. "how things are done here") and inadequate feedback (e.g.,
reporting of incidents) on safety critical processes.
Although we cannot predict when an untoward event can happen, we can say whether it is likely or not. It can be done "by
characterising the variability within the system, specifically the variability of components and subsystems and how they may
combine in unwanted ways. This can be done by looking at how functions and subsystems depend on each other." (Hollnagel,
2016, p. 172). This very information is obtainable from a systemic hazard analysis (Section 4.2) where flawed interactions
between system components at various levels of abstraction are revealed.
These flawed interactions show how safety control can be lost due to inability to adequately deal with operational complexity,
i.e. the inability to anticipate and adjust. The presence of such interactions, or hazardous scenarios, reflects the presence of
variability in the system, and this variability is both necessary and unwanted. The more scenarios to a system hazard, the higher
potential for the unwanted (uncontrolled) variability and also for the needed flexibility to maximise the performance. As
Hollnagel stated "The adaptability and flexibility of human work is the reason for its efficiency. At the same time it is also the
reason for the failures that occur, although it is never the cause of the failures." (Hollnagel, 2016, p. 150). In other words,
"failures are the flip side of success, meaning that there is no need to evoke special failure mechanisms to explain the former.
Instead, they both have their origin in system variability on the individual and systemic levels, the difference being how well the
system was controlled." (Hollnagel et al., 2007, p. xi). Thus, the system variability revealed via a systemic hazard analysis
will lead to hazards if uncontrolled and to successes otherwise.
Thus, the presence of potentially unwanted variability (i.e. found through a hazard analysis) should be regarded as invaluable
information for improving both safety and performance. That is, such system variability indicates the need for better or more
of (Hollnagel and Woods, 2005):

Wind Energ. Sci. Discuss., 2020, 1-20, 10.5194/wes-2020-15, 2020.

• Foresight so hazardous situations can be better anticipated that require speedy adjustments or extra resources (people,
knowledge, competence, time, speed, power, etc.).
• Feedback on physical processes with potential for hazards.
• Flexibility (ability to adjust and adapt) on the part of operational procedures, along with corresponding training of
people;
• Timely availability of safety resources (time, knowledge, competence, etc.) to support adequate adjustment;

Thus, once we have results from a systemic hazard analysis in the form of hazardous scenarios, how do we use them to discern
the existing variability within the system? We propose to use a simple indicator of system variability for a preliminary
comparison of various phases of operation. It is a ratio of the number of hazardous scenarios per operational phase, $\text{NHS}_i$ , to
the total number of hazardous scenarios across all $N$ phases of operation, Eq. (1).

$$\text{System variabilty}_i = \frac{\text{NHS}_i}{\sum_i^N \text{NHS}_i} \qquad (1)$$

Inspired by the Rasmussen's the boundary of safe behaviour (Rasmussen, 1997) and production pressures pushing towards
this boundary, Figure 2 illustrate the rationale behind this indicator of system variability. As discussed above, a systemic hazard
analysis delivers scenarios of interaction between various system components, the interactions that can lead to hazards. These
paths are reflective of the physical design and the majority of them would be constrained by operational procedures in place.
However, under production pressures and performance optimisation, the constraints can get violated and loopholes exploited,
unwittingly drifting to hazardous states as discussed earlier. Figure 2 shows that if hazard 1 and 2 were to belong to different
phases of operation, the first phase would engage more complex operations (because of more complex functions and them
implementing systems) than the second phase, leading to higher variability as a result.

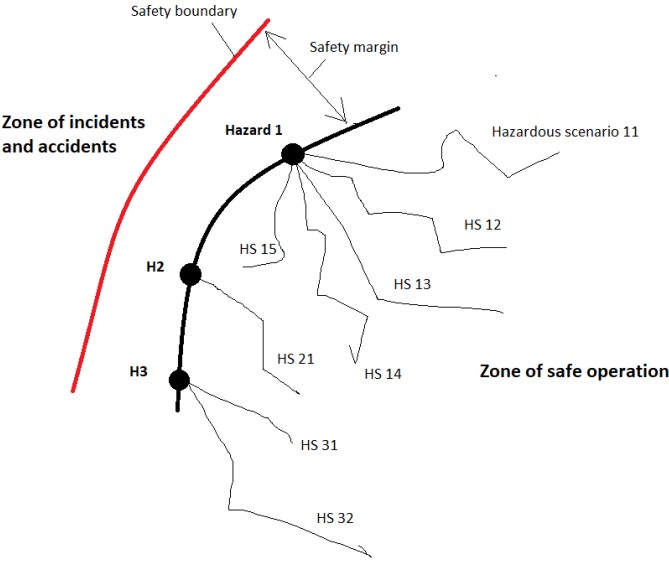

*Figure 2: Graphical interpretation of system variability in the context of safety boundary*

We note that the proposed indictor of system variability does not include the number of hazards the scenarios led to. This is because we are more concerned about the underlying system structure than the events it can produce. This is a more systemic view on the problem; fundamental discussions on this are found in (Meadows, 2008). This also makes the variability indicator less reflective of the number and type of hazards analysed, which can significantly vary from analysis to analysis.

Nevertheless, we recognise that the proposed indicator is not fully independent from how a hazard analysis is performed. Different analysts will produce different set of results for the same system, and hence the system variability will also be different. Therefore, such indicators should not be used to compare different analyses—unless those analyses used the same assumptions—and should be limited to a comparison of different operational scenarios or phases within a single analysis.

The following section introduces to a systemic hazard analysis and its results which are then use compare the operational phases in terms of system variability.

**4.2    Hazard analysis**

### 4.2.1    Systemic accident model

Depending on an accident model assumed, the search principles and analysis goals will be different. Three types accident models can be considered (Hollnagel, 2016): sequential, epidemiological, and systemic. Systemic accident models focus on tight couplings and complex interactions between system components, with the ultimate objective to control the system

variability (Hollnagel, 2016). Such accident models are called systemic because the models require to analyse the system as a
whole, as oppose to analysing its individual components with the purpose to understand the system's behaviour. The explicit
assumption behind systemic accident models is that interactions between system components are more important than
component themselves (Meadows, 2008). Individual components can fail, but if interactions remain adequate, the system will
heal itself, i.e. such failures would be timely detected and mitigated. Notably, system components change all the time (e.g.,
physical components age and get replaced, people come and go), but as long as interactions between them remain the same,
the system fulfils its original purpose.

### 215  4.2.2  STPA

In view of several systemic hazard analysis methods available, we selected the Systems Theoretic Process Analysis
(STPA)(Leveson, 2011a;Leveson and Thomas, 2018). The method is based on systemic accident model STAMP (System-
Theoretic Accident Model and Processes), which is designed for complex, highly automated, socio-technical systems
(Leveson, 2004;Leveson, 2011b). The comparison of STPA and STAMP with other analysis methods and accident models can
be found in the literature, e.g.(Salmon et al., 2012;Sulaman et al., 2019;Qureshi, 2007), and it is, hence, disregarded in this
paper.
Before explaining the method, it would be helpful to agree on the terminology used. A hazard is a system state that will lead
to an incident or accident given specific environmental conditions beyond the control of system designer (Leveson, 2004). The
system in question can be a safety management system (SMS) which is designed according to the ISM Code or amalgamated
from different RAMS. Incidents and accidents are defined as follows (Rausand, 2013). An incident is a materialised hazard
with insignificant consequences. Incidents do not necessary interrupt the prime function (delivery of payload or service). An
accident is a materialised hazard with significant consequences (significant loss or damage). Accidents would normally
interrupt the prime function.

Wind Energ. Sci. Discuss., 2020, 1-20, 10.5194/wes-2020-15, 2020.

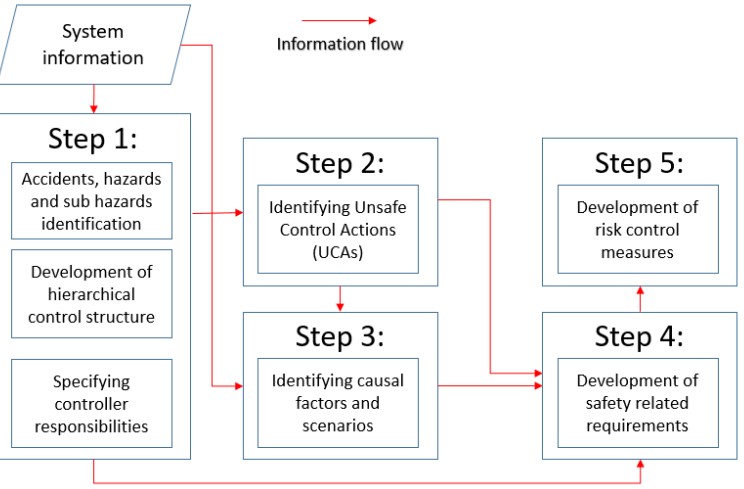


*Figure 3: STPA process*
A sequential process behind the STPA method is shown in Figure 3. The analysis begins by defining the system and its
boundaries. This allow clarifying what accidents (losses) and system-level hazards (conditions for incidents) should be
considered in the analysis. For instance, during the SOV interface with the turbine via a gangway, the assumed accidents
corresponded to the deviation from the interfacing objective, i.e. occurrence of injuries and life losses, and damages to SOV,
gangway, or turbine. However, the reference to accidents is beyond the scope of this paper, as explained earlier.
Sample system-level hazards are:
1. Vessel does not keep a min safe distance to turbine or its blades (approaching/staying at turbine when it is in motion);
2. SOV does not keep position/heading within target limits for a predefined time;
3. SOV operates on DP class 1, i.e. no redundancy in thrusters, power generation and other safety critical components;
4. SOV transfers technicians when the gangway is disconnected or dysfunctional (e.g., not motion compensated).
The system-level hazards are typically found in safety rules and regulations. The hazards can be further decomposed into (or
described through) sub-system and component-level hazards, which are often more helpful during the analysis. For instance,
the second hazard is equivalent to a situation when DP operational requirements do not request a DP operator to enable DP
class 2 before starting the transfer.
The system definition further involves its modelling as a hierarchical control diagram. It is a natural way to represent many
systems, including safety governance, that involve feedback loops. Figure 4 shows a control diagram for the interface between
SOV and a turbine. The control diagram is at higher level of abstraction, where one controller box comprises three other

Wind Energ. Sci. Discuss., 2020, 1-20, 10.5194/wes-2020-15, 2020.

controllers and controlees: turbine, gangway and technicians being transferred. The arrows indicate control and feedback
channels with example control actions and feedback signals indicated. The control actions reflect the responsibilities assigned
to a controller. The responsibilities, or purpose, are also reflected in the control algorithm and feedback information necessary
for adequate control.

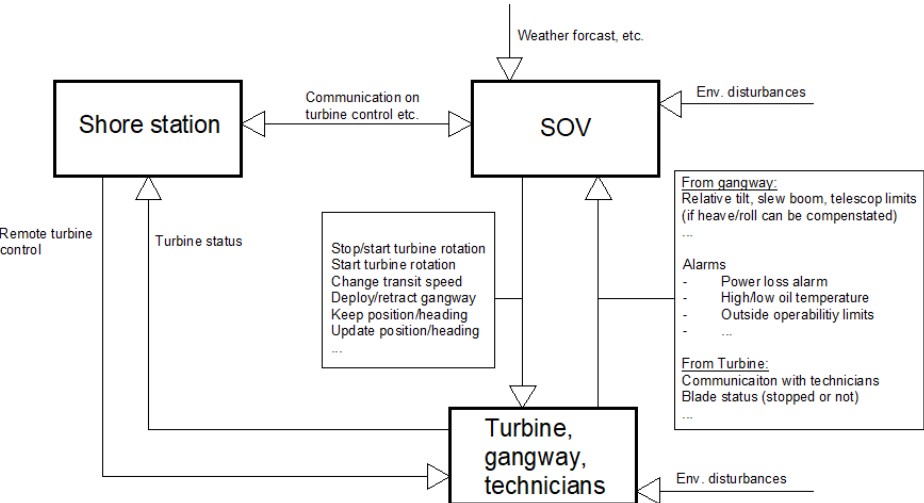


*Figure 4: Hierarchical safety control diagram of interface between SOV and turbine (further explained in Section 4.3)*

The use of control diagram for hazard analysis contrasts with classic analysis methods that instead use failure diagrams such
as fault trees and event trees. The key difference between control and failure diagrams is that the latter show imaginary linear
chains of causes and effects (BS EN 31010:2010). The chains are typically based on past accidents, assuming that future ones
should happen in a similar fashion. The control diagram, on the other hand, does not make such assumptions and shows real
interactions in daily operations. This makes the STPA results credible, easier to communicate and generalise.
The second and third steps of the hazard analysis generate hazardous scenarios, which are then used to develop safety
requirements. A hazardous scenario explains how control actions—from each controller in the control diagram—can lead to
sub-system or system-level hazards, and why this can happen. Scenarios are inferred by searching the operational context (or
states of operation), looking for circumstances—within the entire system—under which a given control action would lead to
a hazard. The STPA uses specific keywords to guide the search (Leveson and Thomas, 2018).

Wind Energ. Sci. Discuss., 2020, 1-20, 10.5194/wes-2020-15, 2020.

The fourth and fifth steps of the hazard analysis in Figure 3 are outside the scope of this paper. However, we provide an
example analysis result which also includes proposed functional requirements. Thus, Figure 5 shows sample hazardous
scenarios and safety requirements for the control action "stop turbine rotation" by SOV controller. The arrows indicate the
scenario as a pathway from basis causal factors to system-level hazards: causal factors cause unsafe control actions, which, in
turn, lead to hazards. The shaded cells illustrate a specific scenario, which is preventable by implementing the three functional
safety requirements. These requirements are complementary, representing organisational and design controls.

| Hazard | Unsafe control actions | Causal factors | Functional requirements |
|---|---|---|---|
| Vessel does not keep a min safe distance to turbine or its blades | Not stopping turbine prior to approaching it | Inadequate communication with the site manager leads vessel operator to wrongly believe the site manager is in control (in reality vessel operator is) of the nacelle and will stop the turbine in time. | Effective communication between the site operator and vessel operator shall be established and maintained |
| | | | When turbines are to be approached for maintenance, the site and vessel operators shall be able to follow the communication procedures |
| | | | When turbines are to be approached for maintenance, SOV control panel (or other design features) shall indicate who is in control of turbine (site manager or vessel) |
| | | Vessel operator wrongly assumes (based on prior experience) the site manager is by default in control of the nacelle and will stop the turbine in time. However, the default situation is opposite - vessel operator is in control unless it is changed | … |
| | | Remote stopping of turbine does not work as intended, and there is no feedback of unsuccess. Therefore, vessel operator assumes it is successful. | … |
| | Turbine rotation is stopped too late, after vessel violates a safe distance to turbine. | … | … |


*Figure 5: Hazardous scenario with three functional requirements*
**4.3    System description**
The overall system in question is shown in Figure 6. The figure shows the analysed interactions between system components
at the system level. These interactions are of physical contact (e.g., SOV and turbine), communication via radio (e.g., SOV

Wind Energ. Sci. Discuss., 2020, 1-20, 10.5194/wes-2020-15, 2020.

and shore, turbine and shore), and sensory (distance, visual, and audio) by installed sensors and people. Other interactions at
the system level (e.g., the links between the DC and turbine or other ships) were not analysed.

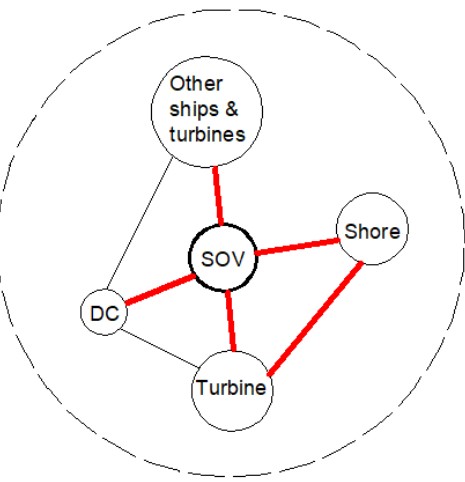


*Figure 6: System components and system boundary (SOV – service operation vessel, DC – daughter craft)*

The considered interactions corresponded to four operational phases:
•    Transit and manoeuvre within a wind farm. The dynamic positioning (DP) system was considered as the main system
providing the navigation and station keeping (position and heading) functions.  The DP system can be switched into
an automatic mode to fully control all three degrees of freedom (DoF): surge, sway, and yaw. The control of DoF can
also be shared with a DP operator who can use a joystick or manual thruster levers.
•    Interface between a SOV and turbine (approach, station keeping, and departure). The DP and motion-compensated
gangway systems were considered to be jointly used. The gangway system is used for technician transfer from SOV
to/from a wind turbine. At the time of transfer, the SOV keeps position and heading by means of the DP system. The
gangway is controlled by a gangway operator who extends, retracts, and maintains communication with the
technicians. There is also a continuous communication between the DP and gangway operators to maintain the
gangway operation within its operability limits.
•    Interface between a SOV and daughter crafts (DC) with a conventional davit system. The DC would be vertically
attached to the davit via a lifting line (vertical) and the painter line to keep the DC aligned with SOV. Both lines are
typically connected and disconnected manually by DC deck crew. DCs are loaded with technicians and equipment,
and launched from a SOV deck by the davit (typically 3-5 times per day) and then recover (lift up) DCs from the
water the same way. During the DC launch and recovery, SOV uses the DP system to maintain the position and

Wind Energ. Sci. Discuss., 2020, 1-20, 10.5194/wes-2020-15, 2020.

heading. The interface between a SOV and DC was assumed to following sub-phases with corresponding systems

and hazards involved: (1) launch from the SOV and recovery of a DC from water using the davit system, (2) and

technician and equipment are transfer when a DC is on water, with technicians claiming up/down the ladder.

These phases of operation are safety critical and there are different safety hazards to watch for (next section). For instance,
during a transit or manoeuvring, the vessel might collide with turbines or other vessels, e.g. when the vessel deviates from a
correct trajectory or inadequately performs collision avoidance.
For each phase, a safety control diagram was developed, e.g. Figure 4 shows the one used for the interface between a SOV
and a wind turbine. Thus, the safety control diagram in Figure 4 was developed by assuming the SOV to be the main controller,
which comprises human controllers on the bridge (e.g., a DP operator), automation, and other ship systems. The shore station
as a controller was not analysed, and only the communication with the SOV was considered. The text next to the arrows explain
their meaning, i.e. what control and feedback information was assumed. The SOV as a controller is generally responsible for
(1) keeping the station (position and heading) until the transfer of technicians via the gangway is complete and (2) providing
power to the gangway. Additionally, it was assumed that these responsibilities are only exercised when the SOV, gangway
and other systems are fully operational. Based on this information, control actions and feedback can be inferred. Technical
publications, such as DP operational manuals, were also used determine control actions and feedback signals (e.g., distance
sensors, GPS signals). As Figure 4 shows, the process under control comprised the gangway and turbine, with controlled
parameters such as the relative distance, bearing, power supply and others.
This phase of SOV operation additionally included a separate hazard analysis of the gangway control, as shown in Figure 7.
The control diagram was developed to reflect industrial safety and other requirements for gangways and technician transfer,
i.e. (IMCA, 2014;DNVGL, 2017, 2015a). The continuous lines correspond to control channels, with the text indicating the
control actions, and dashed lines corresponding to feedback channels. In this diagram, the human operator corresponded to the
gangway operator controlling the gangway position and motions by means of the gangway control system. There is also
communication with technicians who walk via the gangway.

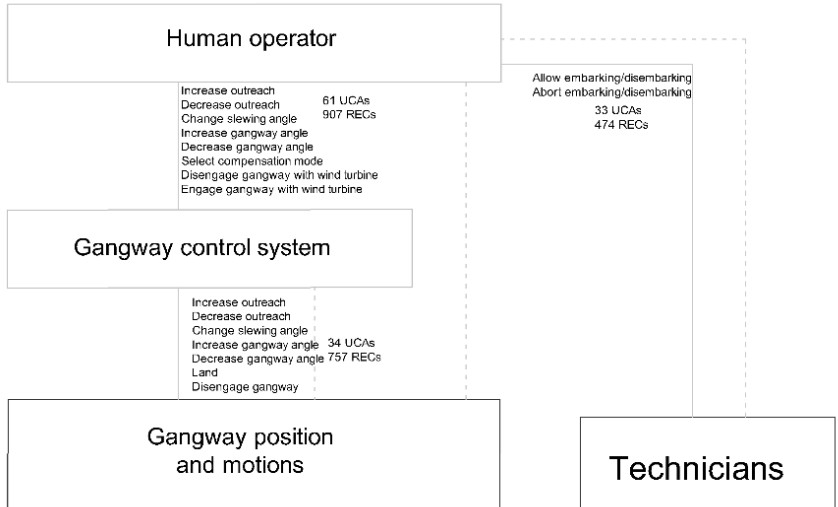


*Figure 7 Gangway control diagram*

Detailed explanations of other control diagrams corresponding to other phases of SOV operation are outwith the scope of this paper. An interested reader is referred to other authors' publications where, for example, a system description and hazard analysis for the DP system in the above phases of SOV operation can be found (Puisa et al., 2019). We note that the safety control diagrams developed for each operational phase were of the same level of abstraction. This makes them comparable, as done in the following section.

**5    Results**

This section outlines the results of hazard analysis by STPA, covering the three stages of SOV operation (Section 4.3). Table 2 to Table 4 outlines the considered hazards, the number of identified scenarios that can lead to them, along with example scenarios meant to demonstrate the interactions involved. As discussed earlier, the number of hazardous scenarios behind a hazard does not automatically mean a higher likelihood for that hazard. Instead, it means that there is a higher system variability, i.e. complexity, in connection with this hazard. If this variability is uncontrolled, it can lead to the hazard.

Based on this tables, Figure 8 shows the system variability as described in Section 4.1. The values indicate that the interface between the SOV and gangway has, potentially, the highest variability. Although, the system variability for the transit and manoeuvring phase is almost the same. The lowest variability is of the SOV interface with daughter crafts.

Wind Energ. Sci. Discuss., 2020, 1-20, 10.5194/wes-2020-15, 2020.

*Table 2: Analysed hazards and their hazard exposure (number of scenarios to hazard) for SOV operational stage: Transit and manoeuvring*

| # | Hazards | Number of scenarios | Example scenarios |
|---|---------|---------------------|-------------------|
| 1 | Thruster control actions mismatch the current mode of operation (i.e. mode confusion) | 259 | Setpoint is not updated when vessel position, heading or trajectory exceeds alarm/alert limits. This can happen when the DP system does not accept new joystick setpoints when the previous task is not yet finished (i.e. the old setpoint has not been yet achieved). |
| 2 | Vessel control actions are in conflict with operational objectives (e.g., position/heading is kept or selected not according to the plan) | 174 | New operational objectives (e.g. move to another position, heading, waypoint) are inadequately (clearly, accurately and timely) communicated and the DP operator does not update the setpoints. |
| 3 | Operation does not comply with the required IMO DP class. This means redundancy against failure of critical components such as thrusters is unavailable. | 11 | When operational objective/circumstances change, operator unwittingly mismatch the DP class to given operational circumstances and does not receive any indicator of the error. |
| 4 | Untimely transfer of thruster control between bridge and engine control room (i.e. inadequate internal communication) | 8 | Because of emergency, crew is distracted or unable to perform a prompt transfer of control. |












Wind Energ. Sci. Discuss., 2020, 1-20, 10.5194/wes-2020-15, 2020.



*Table 3: Analysed hazards and their hazard exposure (number of scenarios to hazard) for SOV operational stage: Interface turbine via gangway*

| # | Hazards | Number of scenarios | Example scenarios |
|---|---------|---------------------|-------------------|
| 1 | Significant gangway motions while personnel (technicians) are on the gangway. Or, gangway structure under increased expansion or compression force as a result of out-of-range gangway/vessel movements. | 169 | Sluggish compensation of relative vertical motions between the SOV and turbine. This can happen due to inadequate predictions of vessel motions or undetected mechanical malfunctions of the gangway. |
| 2 | Vessel does not keep relative position/heading within target limits | 80 | Distance to turbine is not queried when vessel is settling at or keeping the target position as operator does not switch on the distance querying to turbine. |
| 3 | Vessel does not keep a minimum safe distance to the turbine or its blades (incl. vessel approaching a rotating turbine or the turbine starts rotating when the vessel is nearby) | 70 | When the DP/auto mode of approach to turbine is used, manually entered position/heading at the turbine violates the safe distance: typo, wrongly communicated or determined, etc. |
| 4 | Technicians are transferred when the gangway is improperly connected or dysfunctional (e.g., motion compensation is faulty or cannot compensate) | 53 | Deployment of gangway when gangway alarms are active (high oil temp, low oil level, etc.). Given previous experience and management/time pressure, the vessel or gangway operator wrongly assumes that gangway limits are too conservative and alarms are false and it is possible to safely perform the transfer in given env. conditions. |
| 5 | Personnel hands or legs caught between gangway moving parts or between gangway and wind turbine | 50 | The gangway transfer is carried out during bad visibility or external disturbances (e.g., sudden wind, rain, snow). |
| 6 | Gangway is retracted when technicians are being transferred | 26 | Gangway operator reacts mechanically when gangway alarms unexpectedly go off (gangway suddenly reaches the operability limits). |
| 7 | Vessel does not supply required power to gangway continuously | 17 | The vessel operator (and gangway operator) does not check the available power before deploying the gangway. This can happen due to time pressure or inadequate training. |
| 8 | Vessel does not operate on DP class 2 or above. This means redundancy against failure of critical components such as thrusters is unavailable. | 9 | Vessel operator switches on DP2/3 and assumes it is on. However, DP2/3 is not activated due graceful faults or unavailable redundancy (e.g., insufficient power). Meanwhile, operator is busy with other tasks and does not notice. |







*Table 4: Analysed hazards and their hazard exposure (number of scenarios to hazard) for SOV operational stage: Interface with daughter crafts*

| # | Hazards | Number of scenarios | Example scenarios |
|---|---------|---------------------|-------------------|
| 1 | Daughter craft (DC) develops swing or/and spinning motions during launch/recovery | 78 | Securing of DC is inadequately checked before launch/recovery as checking is inconvenient/inhibited due to design features. |
| 2 | Davit does not keep the daughter craft (DC) secured while launching/recovering | 77 | David operator (DO) mechanically switches off davit while launching/recovering DC (only relevant if DC securing can be lost upon switching off davit) as DO receives "abort" order from the bridge / other crew members. |
| 3 | Daughter craft (DC) develops excessive motions on water when being launched or about to be recovered | 42 | David operator (DO) starts launch of DC during excessive waves/current. This can happen when DO mechanically follows orders from an uninformed coordinating officer. |
| 4 | SOV interfaces with the daughter craft (DC) when SOV is unable to maintain position/heading (either automatically or manually) | 38 | SOV bridge operator does not wait until the DP settles before the DC launch can proceed. This can be because of time pressure, lack of training, or lack of feedback on the DP settlement status. |
| 5 | Davit violates the maximum launching speed of the daughter craft (DC), leading to damage caused by impact on water | 25 | David operator (DO) starts launch of DC when SOV is at speed or the SOV speed increases during the time of DC launch. |
| 6 | Technicians moving on the SOV ladder are unsecured (unprotected from falls, trips, and slips). | 21 | Despite significant motions (accelerations) of SOV, technician wrongly assumes it is ok to use just one hand while climbing the ladder. |
| 7 | While on the SOV or water, daughter craft (DC) abruptly shifts when technicians getting in/out DC or when DC crew is working on deck | 17 | Davit Operator (DO) retracts davit lines when DC is still being detached by DC crew. DO underestimates the time needed to detach DC and communicates it to DO before completing the task. This scenario can happen due to time pressure, or ignorance of environmental conditions that can prolong the task. |
| 8 | SOV interfaces with the daughter craft (DC) when either of ships experience excessive motions | 16 | Due to delayed forecast of env. conditions, the SOV bridge permits the DC launch in environmental conditions which quickly deteriorate during the launch. |
| 9 | Technicians are crossing from SOV ladder to/from the daughter craft (DC) when a gap between SOV and DC is too big or increasing (DC is not pushing against SOV). | 12 | Technician steps over without waiting (immediately) until DC starts pushing against SOV. This can happen because the crossing process is not coordinated by a safety officer or coordinated inadequately. |
| 10 | Horizontal centre-of-gravity of the daughter craft (DC) is significantly misaligned with respect to the lifting hook line. | 11 | Correctness of DC loading is inadequately checked before launching DC, because david operator (or other crew) does not have adequate skills/knowledge or checking was impeded. |
| 11 | Technicians are crossing from the SOV ladder to the daughter craft (DC) too slowly. | 7 | Technician are unaware that crossing should be instant: unfamiliar with safety instructions or the crossing is inadequately coordinated. |


Wind Energ. Sci. Discuss., 2020, 1-20, 10.5194/wes-2020-15, 2020.

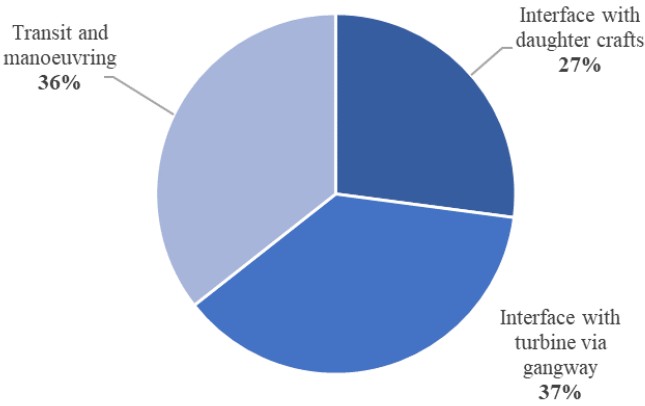

*Figure 8: System variability for the three stage of SOV operation*

## 6    Discussion

The presented results of the systemic hazard analysis are twofold. First, they bring awareness of system-level hazards involved in various stages of SOV operations, although the existing industrial rules and good practices are likely to cover them. For instance, the sample scenario for the hazard in Table 3 "Vessel does not keep a minimum safe distance to the turbine or its blades" is addressed by class rules which require the DP system to perform self-check routines and bring the system to a stop if necessary (DNVGL, 2015b). However, these technical publications do not explain how the rules or guidelines can be violated and what the level of complexity involved when following them.

This bring us to the second contribution of the study, namely the exposure of the variability in the system when in various phases of operation. All three phases of SOV operation have rather comparable levels of system variability. However, the interface between the SOV and turbine via the gangway system and the manoeuvring between turbines seem to be more complex phases of operation where the potentially uncontrolled variability is more likely. The similarity between these two phases may come from the fact that the DP system is used in both of them, and this system is quite complex. At the same time, the gangway system does not seem to add a significant amount of variability in the analysis we have performed.

As argued in Section 4.1, the variability in the system is inevitable and necessary to achieve its goals in view of internal deficiencies and external disturbances. However, the variability should be controlled. Uncontrolled variability means that necessary adjustments to keep the system healthy are not performed or performed inadequately due to lack of resources, capabilities, or other reasons. The results of the systemic hazard analysis can be used to locate where such uncontrolled variability is more likely—the likelihood arguably increases with the number of hazardous scenarios—and where resources and capabilities should be added to facilitate performance adjustments. Section 4.1 has already highlighted what general

resources and capabilities should be considered to improve the ability to anticipate, adjust and better control the system
variability.
A detailed discussion on what specific resources and capabilities should be added to each phase of SOV operation is outside
the scope of this paper. The presented study only shows that such changes should begin with the SOV interface with turbines,
including the approach and departure. The improvements will aim to avoid untoward events such as the earlier highlighted
accident with Vos Stone in Section 1.
**7    Limitations**
The proposed indicator of the system variability is only suitable for some preliminary analysis. The paper has not validated
the indicator by analytical or empirical means. However, the presented theoretical basis and used assumptions therein provide
a reasonable support for the indictor. Clearly, further research is needed in this still new area of systemic safety analysis.
**8    Conclusions**
The paper has presented the results of systemic hazard analysis of service offshore vessel's (SOV) operations. We have
specifically analysed 23 operational hazards arising during the three stages of SOV operation: (1) transit and manoeuvre within
a windfarm and interfaces with (2) turbines and (3) daughter crafts. The hazards are mostly related to flawed interactions
between people and technology, as opposed to individual failures (e.g., human errors, random failures of equipment) that are
addressed conventionally. During the hazard analysis, we identified 1,270 hazardous scenarios that explain how hazards can
materialise.
The study has made the following contributions:
• It has brought awareness of system-level hazards involved in various phases of SOV operation and the number of
hazardous scenarios associated with them. In this study, the number of hazardous scenarios is not interpreted as being
proportional to the hazard likelihood (i.e., probability or frequency). Instead, it is interpreted as the degree of system
variability which, on the one hand, is necessary for normal performance and, on the other hand, is indicative of system
complexity and the potential for uncontrolled variability. We broadly discussed what resources and capabilities should
be added to improve the control. Knowledge where the system variability is highest, gives an opportunity to improve
both performance (efficiency) and safety. In the latter case, improvements can be introduced to risk assessments (e.g.,
by updating assumptions), RAMS (or hazard logs), safety procedures, and training of new and existing operations.
• The paper has compared the operational phases in terms of a systemic indicator–the system variability—across three
phases of SOV operation. All three phases of SOV operation have rather comparable levels of system variability.
However, the interface between the SOV and turbine via the gangway system and the manoeuvring between turbines
seem to be more complex, exhibiting higher variability within the system. The comparison can be seen as an
alternative to conventional comparisons where qualitative or quantitative, typically aggregative, figures are used. As
argued in the paper, the aggregation of hazards or hazardous scenarios into system-level indices is an
oversimplification.
• The study has also shown how results of a systemic hazard analysis can be linked to systemic indicators or measures
of system safety, namely to the system variability.

## 9    Acknowledgement

The work described in this paper was produced in research project NEXUS (https://www.nexus-project.eu). The project has

received funding from the European Union's Horizon 2020 research and innovation programme under agreement No 774519.

The authors are thankful to their colleagues and project partners who directly and indirectly contributed to the presented work.

Particular thanks go to Kongsberg Maritime (former Rolls Royce Marine) for sharing design information and providing

valuable feedback. The sponsorship of the Maritime Research Centre by DNV GL and Royal Caribbean Cruises Ltd is also

much appreciated.

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
