# Peer review of "Revealing system variability of offshore service operations through systemic hazard analysis"

_Wind Energy Science, 2020_

## Referee Comment (RC1) · Anonymous Referee #1 · 16 Apr 2020

The submitted manuscript touches upon relevant subject of safety assurance of a complex socio-technical system. The manuscript is properly structured and utilizes appropriate methods and data to achieve its goal, which is the development of a tool arising the awareness of hazards that may have been overlooked in earlier assessments of the system.

The manuscript offers good reading, and deserves publication, subject to some minor issues, of rather technical nature.

Line 210 - it is not clear to what specific steps of hazards analysis the authors are referring. Some graphical representation of this process would be helpful.

Tables 3-5 Since one hazard may results in numerous scenarios (e.g. hazard #1 creates 259 scenarios), please elaborate more on the process of transition from a hazard to a scenario. It is not clear at this stage.

Figure 6 Please explain the meaning of relative exposure to hazards and the way this index is calculated.

---

## Author Comment (AC1) · 20 Apr 2020

Referee comments:

Comment1: Line 210 - it is not clear to what specific steps of hazards analysis the authors are referring. Some graphical representation of this process would be helpful. Comment2: Tables 3-5 Since one hazard may results in numerous scenarios (e.g. hazard #1 creates 259 scenarios), please elaborate more on the process of transition from a hazard to a scenario. It is not clear at this stage. Comment3: Figure 6 Please explain the meaning of relative exposure to hazards and the way this index is calculated.

Author response: All comments are valid and the manuscript has been updated appropriately. Comment1: A cross-reference to Fig.2 has been inserted where the steps

of hazard analysis are given. Comment2: Section 5 now contains explanation about the transition between hazards and their scenarios. Comment3: Section 4.4 has been updated to better explain the relative exposure.

---

## Referee Comment (RC2) · Anonymous Referee #2 · 10 Jul 2020

General comments:
The authors points out that SOV operations constitute a complex socio-technical system, consisting of several subsystems. Each subsystem has specified safety procedures, but the authors argue that potential hazards resulting from the interaction of subsystems might not be properly accounted for with the existing approach, which focus on potential hazards for each subsystem. Instead, they propose to use the systemic method STPA, where the starting point is to identify hazards on the system level. The topic of the paper is relevant, and it is generally well structured. However, I see some issues in the quantitative outcome of the analysis, as given below. The authors make conclusions about the risk in the abstract, however, they do not actually do a risk analysis, as they do not consider the consequences/potential losses, and also not

directly the likelihood. They write in line 224-225: "As incident prevention is the focus on this paper, the likelihood alone can be used to rank the hazards, provided the consequences all considered hazards are similarity intolerable." However, there is no justification that the hazards are similarly intolerable. In fact, according to the reference for the applied STPA analysis (Leveson and Thomas 2018), the first step of the analysis should be to identify losses (before hazards are identified). However, the authors seem to omit this part of the analysis. I suggest the authors to either modify the analysis to include the losses, or to clearly state in the paper and abstract that this part of the analysis is omitted,and not to make conclusions on the risk.

The hazard exposure is quantified through the number of scenarios leading to each hazard, and results are presented in tables 3-5. It is not clear to me, whether this is this the number of different scenarios, or if it is some frequency of exposure. It is also not clear to me how these scenarios were actually found – e.g. were they found using some documents, in dialog with Kongsberg Maritime, using incident reports, or other? If the number of scenarios is to be understood as the number of different scenarios that can potentially lead to a hazard, I do not follow the argument, that it can be seen as a measure of exposure. The exposure would depend on the probability/frequency and duration of the "scenarios". The number of scenarios seem to depend mainly on how things are defined, and how exhaustive/imaginative the analyst is. The assumption that the hazard exposure can quantified through the number of scenarios leading to each hazard seems to be made, in order to come up with a quantitative outcome of the analysis, but I cannot see the need (or justification) for coming up with a quantitative outcome. In the abstract, it is written: "The objective of this paper is to bring awareness of hazards that may have been overlooked in earlier assessments, and allow for a preliminary comparison of various operational stages." Specifically, they wish to "explore hazardous scenarios caused by flawed interactions between system components". Based on this, I believe that the main outcome of the analysis, and the main result to be presented in the paper, should be the identified "hazardous scenarios caused by flawed interactions between system components" If such scenarios were

not found, the abstract and conclusion should reflect this.

Specific comments:
Line 32-33: "It is normally a motion-compensated (3 or 6 DoF) gangway system, which allows for relatively safer (based on experience so far) and time-efficient (within some 5 minutes) transfer." I believe that the bump-and-jump method is significantly more time-efficient, if conditions allow for transfer this way - this could be mentioned. (see e.g. Nikki Twigt, Access Systems for Offshore Turbines - A review of conventional and walk-to-work transfer methods, 2020)

Technical corrections:
Line 72: Reference to Section 0
The paper needs a language check.

―――――――――――――――――――

---

## Referee Comment (RC3) · Anonymous Referee #3 · 27 Jul 2020

Major comments:

1. About the objective, as stated in abstract and line 60, the objective of the paper is 1) to bring awareness of hazards that may have not been captured in earlier assessments, and 2) allow for a preliminary comparison of various operational stages of SOV. The second objective is clearly addressed in the paper. However, the first one: capturing hazards that may have been overlooked by earlier assessments, the so-called earlier assessments are not specific enough in the paper. There is no way to judge whether those identified hazards were overlooked in earlier assessments. The only place that mentioned earlier assessments is part 2 related work. However, the review work does not seem to be complete or provide enough evidence that identified hazards in this work is more complete than others. My arguments are 1) defined scope of hazard

analysis are different, therefore resulting different hazards identified, for example, both Presencia and Shafiee, Dong et al. focused on collision, Rokseth et al. focused on DP system. 2) The depths of hazard analysis are different. The definition of hazards is relative, for example, the occurrence of an event can be considered as an accident in one analysis however can be defined as a hazard in another analysis. We can always find root causes after root cause if we dive deeper. If the authors want to compare with earlier assessments, you need to compare with the assessment which considered the same system or scope of analysis.

I suggest the authors delete "that may have not been captured in earlier assessments" from your objective.

2. System description. There is no description about the system to be analyzed in the work. To conduct hazard analysis by STPA, the first step is to define system. However, the author did not clearly specify the system boundaries and components included, even though the authors described the different phases of operation in 4.1.

In Figure 3, shore station is included, however the control loops between shore station and SOV, and shore station and turbine are not part of your analysis. This is fine. It could be better to mark that shore station is not within your defined system scope.

The authors did not specify what accident (unwanted losses) were considered in the analysis either. This is the very important part when applying STPA method. Using "For instance. . ." in Line 176 may not be enough to specify targeted accidents (Loss). Especially the authors wanted to focus on "to bring awareness of hazards that may have not been captured in earlier assessments". Without being specific, readers need to assume that your work covered all accidents and incidents, such as a heart attack of a personnel during operation. The example I gave here is bit too extreme but is within the definition of accident (unexpected and unwanted loss of life).

3. Comparison of different operational phases. There are several issues here. First, the authors chose median value for comparison without providing argument to support

that the median value is a proper choice. Second, the authors assumed that 1) all hazards are equally intolerable, and 2) number of hazardous scenarios present the occurrence probability. These two assumptions are not widely accepted. This is fine since the authors clearly stated that those are assumptions. However, the authors did not discuss the inaccuracy of the conclusion due to these assumptions. Third, obviously, there are other quantities that can be used for ranking as well. The ultimate objective is to compare different operation phases. If we median value, then the rank (high to low risk) is transit and maneuvering > Interface turbine via gangway > interface with daughter craft. If we use the total number of hazardous scenarios of each phase, then the rank (high to low risk) is Interface turbine via gangway (474 hazardous scenarios) > transit and maneuvering (452) > interface with daughter craft (344). However, if we use the total number of hazards of each phase, then the rank (high to low risk) is interface with daughter craft (11)> Interface turbine via gangway (8) >transit and maneuvering (4). Different quantities can provide different results, and I can make reasonable assumptions to support the rationality of all these quantities.

Since ranking is a focus of the paper as stated in the objective, the authors need to provide enough supportive arguments for their choice and discuss other opposite possibilities as well.

4. Limitations of the method. The authors did not specify the limitations of using STPA methods for hazard analysis. A method is not almighty, limitations of the method applied should be specified as well besides the biases and knowledge limitation from the analyzers.

5. Language check is required. There are many misspellings in the paper, for example, "Specufic control actions" in Figure 4, Line 72 "Section 0".

Minor comments:

1. Line 164-166, about the definition of incident and accident. 1) Even though the authors referred Rausand's differentiation between incidents and accidents, this differentiation is not used anywhere in the paper at all. From my point of view, authors focused on possible accidents. However, incident is the terms showed up in many places in the paper which actually means accidents (if I understand it correctly). 2) If the authors defined incident is a materialized hazard with insignificant consequences, how accident can be incidents with significant consequences?

I suggest the authors either delete this differentiation or be more precise when using these two terms if such a differentiation is made in the paper. In addition, a thorough check of where accident and incident are used is necessary.

2. Line 181, Line 269 Table 3, Line 282, Table 4. "3. SOV does not operate on DP class 2 or above" etc. Therefore, I do not think this can be considered as an operational hazard. Or at least, the way the authors phrased it does not seem right. DP failure sounds more like a hazard.

3. Line 208, Figure 4. The author did not explain why there are black box and white box in the figure. I think Figure 4 can be deleted. The information from Figure 4 is already covered by Figure 4. Figure 4 does not provide any new information from my point of view.

4. Line 293, Table 6. It is a bit strange that during transit and maneuvering phrase, there is no inadequacy in control actions & feedback and/or process model. A simple scenario can be the collision accident between installed turbine (if there are some other turbines which have been installed in the field) and vessel, or vessel and ice. A possible hazard is "the vessel does not keep a min safe distance to turbine or its blades. Possible hazardous scenario can be that the vessel stops too slowly (especially in manual navigation), turbine(blade) is not noticed etc. The same applies to that there is no inadequacy in handling external disturbances during interface with turbine via gangway and interface with daughter craft.

5. The second and third paragraph in the discussion part. Well, the appearance of these two paragraphs is a bit abrupt. In addition, it does not highlight the necessary/advantage of conducting such a hazard analysis as the authors did. The authors first argued that analyzed hazards should be already covered by existing safety rules or regulations, however it is not guaranteed those rules can be followed in practice. Well, this can be true that rules may not be followed in practice, therefore there is still residual risk during the operation. However, residual risk also exists after a hazard analysis (risk analysis) is conducted. Hazards are identified does not mean that hazards are eliminated. The authors also judged that some hazardous scenarios are not addressed by regulations for certain reasons. However, in my opinion, those reasons also apply to the hazard analysis that the author conducted. Rule/regulations are not perfect, so is your hazard analysis.

If you want to compare the hazards you identified with those stated by rules/regulations, at least you should start with the same defined scope of analysis, the same type of accident (loss), the same hazards.

I suggest that the authors rephrase these two paragraphs or can simply delete them.

6. The first paragraph in the conclusion part. I already stated my argument in the first comment about the objective. There is not enough evidence showing that risk assessments are done piecemeal and potentially lacking completeness. . .. It may sound better if you simply say that no such risk assessment has been conducted in the industry (only if experts from industry told you so). The value of academic work is not to conduct a specific risk assessment, sure you cannot find such from published papers. Conducting hazard analysis/risk assessment is an industrial practice which do not necessary provide enough academia value.

7. The detailed intermediate results and final results from the STPA can be added in the appendix. For example, the control diagrams, identified unsafe control actions.
* * *

---

## Author Comment (AC2) · 24 Aug 2020

The response to the reviewer is attached as a pdf file. Also pasted here:
General comments: The authors points out that SOV operations constitute a complex socio-technical system, consisting of several subsystems. Each subsystem has specified safety procedures, but the authors argue that potential hazards resulting from the interaction of subsystems might not be properly accounted for with the existing approach, which focus on potential hazards for each subsystem. Instead, they propose to use the systemic method STPA, where the starting point is to identify hazards on

the system level. The topic of the paper is relevant, and it is generally well structured. However, I see some issues in the quantitative outcome of the analysis, as given below. The authors make conclusions about the risk in the abstract, however, they do not actually do a risk analysis, as they do not consider the consequences/potential losses, and also not directly the likelihood. 1. Author's response to the preceding paragraph

Indeed, risks associated with the identified hazards, in terms of likelihood and consequences, are not presented in the paper. The scope of the paper is limited to hazard prevention/control, i.e. hazard mitigation is outside the scope. To make sure the message is unambiguous, appropriate changes will be made in the abstract and elsewhere in the paper.

However, based on other review comments, the paper has updated and doesn't not use the hazard likelihoods either. Instead, the comparison is done by using a systemic indicator – system variability.

They write in line 224-225: "As incident prevention is the focus on this paper, the likelihood alone can be used to rank the hazards, provided the consequences all considered hazards are similarity intolerable." However, there is no justification that the hazards are similarly intolerable. In fact, according to the reference for the applied STPA analysis (Leveson and Thomas 2018), the first step of the analysis should be to identify losses (before hazards are identified). However, the authors seem to omit this part of the analysis. I suggest the authors to either modify the analysis to include the losses, or to clearly state in the paper and abstract that this part of the analysis is omitted, and not to make conclusions on the risk. 2. Author's response to the preceding paragraph

The quotation from the paper refers to worst-case consequences of the hazards, and they are quite certain. As stated in the paper (Section 4.2, lines 176-177), all analysed hazards will lead to the same accidents /consequences: injuries and life losses, and damages to SOV, gangway, or turbine. Because consequences are the same for all hazards, we only rank hazards based on their degree of exposure – the proxy for

likelihood. We agree it should be stated more clearer what accidents/consequences are considered and that all analysed hazards are assumed to lead to them.

However, based on other review comments, the paper has updated and doesn't not use the hazard likelihoods either. Instead, the comparison is done by using a systemic indicator – system variability.

The hazard exposure is quantified through the number of scenarios leading to each hazard, and results are presented in tables 3-5. It is not clear to me, whether this is this the number of different scenarios, or if it is some frequency of exposure. It is also not clear to me how these scenarios were actually found – e.g. were they found using some documents, in dialog with Kongsberg Maritime, using incident reports, or other? If the number of scenarios is to be understood as the number of different scenarios that can potentially lead to a hazard, I do not follow the argument, that it can be seen as a measure of exposure. The exposure would depend on the probability/frequency and duration of the "scenarios". 3. Author's response to the preceding paragraph

In this paper, the exposure refers to the number of scenarios in the lead up to a hazard (see Section 4.4). In other words, it is the number of pathways to a hazardous state. Some of these pathways are addressed by design and management measures, while others may be overlooked or intentionally discounted as being unlikely. Thus, the exposure is not time related in the context of this paper, and the dictionary definition of exposure does not imply the link to time. The dictionary definition of exposure is "the state of being in a place or situation where there is no protection from something harmful or unpleasant" (ref. Oxford English Dictionary).

However, based on other review comments, the paper has updated and doesn't not use neither the hazard likelihood or exposure to hazard anymore.

The number of scenarios seem to depend mainly on how things are defined, and how exhaustive/imaginative the analyst is. 4. Author's response to the preceding paragraph

That is correct. The analysis was performed based on the technical documentations available, and discussions with designers and operators. There is indeed a real possibility that some scenarios were overlooked. However, it is a common practice in safety engineering when the analysis is done manually. Unfortunately, automated analysis that could exhaustively explore all scenarios is unavailable.

As indicated in responses above, we adopted a systemic indicator to be less sensitive to this problem.

The assumption that the hazard exposure can quantified through the number of scenarios leading to each hazard seems to be made, in order to come up with a quantitative outcome of the analysis, but I cannot see the need (or justification) for coming up with a quantitative outcome. In the abstract, it is written: "The objective of this paper is to bring awareness of hazards that may have been overlooked in earlier assessments, and allow for a preliminary comparison of various operational stages." Specifically, they wish to "explore hazardous scenarios caused by flawed interactions between system components". Based on this, I believe that the main outcome of the analysis, and the main result to be presented in the paper, should be the identified "hazardous scenarios caused by flawed interactions between system components" If such scenarios were not found, the abstract and conclusion should reflect this.

5. Author's response to the preceding paragraph

All hazardous scenarios in Tables 3-5 are of interaction nature. The quantification in terms of number of scenarios is not actually done, and it is never claimed that it has been. The exposure (vulnerability) to hazards is used only to give guidance as to what priority of the follow-up, more detailed and potentially quantitative analysis should follow. This is stated on Line 258-259: " The comparison is, nevertheless, preliminary and should be used as a preface for a more detail, potentially quantitative, comparison."

Thus, the paper delivers system level hazards and associated information.

Specific comments: Line 32-33: "It is normally a motion-compensated (3 or 6 DoF) gangway system, which allows for relatively safer (based on experience so far) and time-efficient (within some 5 minutes) transfer." I believe that the bump-and-jump method is significantly more time-efficient, if conditions allow for transfer this way - this could be mentioned. (see e.g. Nikki Twigt, Access Systems for Offshore Turbines - A review of conventional and walk-to-work transfer methods, 2020)

6. Author's response to the preceding paragraph

It is indeed can be more time efficient, but less safer, based on our opinion. We'll familiarise with the publication and cite it if relevant.

Technical corrections: Line 72: Reference to Section 0 The paper needs a language check.

7. Author's response to the preceding paragraph

The corrections will be made. Thank you.

Please also note the supplement to this comment:
https://wes.copernicus.org/preprints/wes-2020-15/wes-2020-15-AC2-supplement.pdf

---

## Author Comment (AC3) · 24 Aug 2020

The response to Reviewer 3 is attached as a pdf, and also pasted here.
to judge whether those identified hazards were overlooked in earlier assessments. The only place that mentioned earlier assessments is part 2 related work. However, the review work does not seem to be complete or provide enough evidence that identified hazards in this work is more complete than others. My arguments are 1) defined scope of hazard analysis are different, therefore resulting different hazards identified, for example, both Presencia and Shafiee, Dong et al. focused on collision, Rokseth et al. focused on DP system. 2) The depths of hazard analysis are different. The definition of hazards is relative, for example, the occurrence of an event can be considered as an accident in one analysis however can be defined as a hazard in another analysis. We can always find root causes after root cause if we dive deeper. If the authors want to compare with earlier assessments, you need to compare with the assessment which considered the same system or scope of analysis. I suggest the authors delete "that may have not been captured in earlier assessments" from your objective. 1. Author's response to the preceding paragraph

Indeed, the paper doesn't provide an explicit, systematic comparison with the state of the art. And so is not claimed in the paper. Such a comparison is only possible if, as rightly pointed by the reviewer, there a hazard analysis conducted in similar settings to compare with. The authors have not come across such an analysis.

This and other comments by the reviewer, prompted us to revise the paper and introduce significant changes in to the objectives/scope as well.

2. System description. There is no description about the system to be analyzed in the work. To conduct hazard analysis by STPA, the first step is to define system. However, the author did not clearly specify the system boundaries and components included, even though the authors described the different phases of operation in 4.1.

2. Author's response to the preceding paragraph

We agree with the comment concerning the insufficient of system description in terms of boundaries and components. The revised version will have more detail on this.
However, the paper will be limited to a high-level description, given that hazardous scenarios are outwith this paper. We also limit the description to one of the operational phases only.

In Figure 3, shore station is included, however the control loops between shore station and SOV, and shore station and turbine are not part of your analysis. This is fine. It could be better to mark that shore station is not within your defined system scope.

3. Author's response to the preceding paragraph

Actually it was a part of the analysis. The analysis was under Hazard 3 in Table 4. We had scenarios of miscommunication between shore and SOV, leading to this hazard. We've added extra text for this hazard.

The authors did not specify what accident (unwanted losses) were considered in the analysis either. This is the very important part when applying STPA method. Using "For instance. . ." in Line 176 may not be enough to specify targeted accidents (Loss). Especially the authors wanted to focus on "to bring awareness of hazards that may have not been captured in earlier assessments". Without being specific, readers need to assume that your work covered all accidents and incidents, such as a heart attack of a personnel during operation. The example I gave here is bit too extreme but is within the definition of accident (unexpected and unwanted loss of life).

4. Author's response to the preceding paragraph

We agree. However, as the revised manuscript shows, this become irrelevant. The study focusing on hazards and their scenarios only. There is no ranking based on risk.

3. Comparison of different operational phases. There are several issues here. First, the authors chose median value for comparison without providing argument to support that the median value is a proper choice.

5. Author's response to the preceding paragraph

The revised manuscript proposes a systemic indicator for this comparison. The median is not used anymore.

Second, the authors assumed that 1) all hazards are equally intolerable, and 2) number of hazardous scenarios present the occurrence probability. These two assumptions are not widely accepted. This is fine since the authors clearly stated that those are assumptions. However, the authors did not discuss the inaccuracy of the conclusion due to these assumptions.

6. Author's response to the preceding paragraph

The revised manuscript (thanks to the comments by this reviewer) has significant changes and does not refer to probabilities and provides a different interpretation of hazardous scenarios.

Third, obviously, there are other quantities that can be used for ranking as well. The ultimate objective is to compare different operation phases. If we median value, then the rank (high to low risk) is transit and maneuvering > Interface turbine via gangway > interface with daughter craft. If we use the total number of hazardous scenarios of each phase, then the rank (high to low risk) is Interface turbine via gangway (474 hazardous scenarios) > transit and maneuvering (452) > interface with daughter craft (344). However, if we use the total number of hazards of each phase, then the rank (high to low risk) is interface with daughter craft (11)> Interface turbine via gangway (8) >transit and maneuvering (4). Different quantities can provide different results, and I can make reasonable assumptions to support the rationality of all these quantities. Since ranking is a focus of the paper as stated in the objective, the authors need to provide enough supportive arguments for their choice and discuss other opposite possibilities as well.

7. Author's response to the preceding paragraph

This is a very good observations. This comment is the main cause for substantial

changes the paper has undergone. We now use a different measure and provide a detailed rationale why it makes sense.

4. Limitations of the method. The authors did not specify the limitations of using STPA methods for hazard analysis. A method is not almighty, limitations of the method applied should be specified as well besides the biases and knowledge limitation from the analyzers.

8. Author's response to the preceding paragraph

References to other publications which address the STPA limitations will be added.

5. Language check is required. There are many misspellings in the paper, for example, "Specufic control actions" in Figure 4, Line 72 "Section 0".

9. Author's response to the preceding paragraph

The language will checked and improved.

Minor comments: 1. Line 164-166, about the definition of incident and accident. 1) Even though the authors referred Rausand's differentiation between incidents and accidents, this differentiation is not used anywhere in the paper at all. From my point of view, authors focused on possible accidents. However, incident is the terms showed up in many places in the paper which actually means accidents (if I understand it correctly). 2) If the authors defined incident is a materialized hazard with insignificant consequences, how accident can be incidents with significant consequences? I suggest the authors either delete this differentiation or be more precise when using these two terms if such a differentiation is made in the paper. In addition, a thorough check of where accident and incident are used is necessary. 10. Author's response to the preceding paragraph

The used understanding is that depending on consequences of an untoward event, it is called a near-miss, incident or accident. Near-misses would not have consequences at all, incidents will have insignificant ones, whereas accidents will have significant

consequences. The term incident is also used to refer all untoward events when con-sequences are assumed unknown, but this is indeed confusing and a stricter definition should be used. Corrections will be made.

2. Line 181, Line 269 Table 3, Line 282, Table 4. "3. SOV does not operate on DP class 2 or above" etc. Therefore, I do not think this can be considered as an operational hazard. Or at least, the way the authors phrased it does not seem right. DP failure sounds more like a hazard.

11. Author's response to the preceding paragraph DP class 2 or higher introduces redundancy against thruster and other component/sub-system failures. It is a safety procedure that has to be followed when the ship is engaged in safety critical tasks. Therefore, operating without redundancy on, represents a hazard. Text will be added to explain all this.

3. Line 208, Figure 4. The author did not explain why there are black box and white box in the figure. I think Figure 4 can be deleted. The information from Figure 4 is already covered by Figure 4. Figure 4 does not provide any new information from my point ofview.

12. Author's response to the preceding paragraph

Figure 4 clarifies the input for the analysis, as indicated on Line 206. But it is indeed redundant and will be removed.

4. Line 293, Table 6. It is a bit strange that during transit and maneuvering phrase, there is no inadequacy in control actions & feedback and/or process model. A simple scenario can be the collision accident between installed turbine (if there are some other turbines which have been installed in the field) and vessel, or vessel and ice. A possible hazard is "the vessel does not keep a min safe distance to turbine or its blades. Possible hazardous scenario can be that the vessel stops too slowly (especially in manual navigation), turbine(blade) is not noticed etc. The same applies to that there

is no inadequacy in handling external disturbances during interface with turbine via gangway and interface with daughter craft.

13. Author's response to the preceding paragraph

The hazard "the vessel does not keep a min safe distance to turbine or its blades" was assigned to the phase when the vessel is interfaced with the turbine. During transit and manoeuvring, other hazards were considered (Table 3) and amongst them, credible causal factors were not related to control and feedback channels. This clarification will be added into the text.

5. The second and third paragraph in the discussion part. Well, the appearance of these two paragraphs is a bit abrupt. In addition, it does not highlight the necessary/advantage of conducting such a hazard analysis as the authors did. The authors first argued that analyzed hazards should be already covered by existing safety rules or regulations, however it is not guaranteed those rules can be followed in practice. Well, this can be true that rules may not be followed in practice, therefore there is still residual risk during the operation. However, residual risk also exists after a hazard analysis (risk analysis) is conducted. Hazards are identified does not mean that hazards are eliminated. The authors also judged that some hazardous scenarios are not addressed by regulations for certain reasons. However, in my opinion, those reasons also apply to the hazard analysis that the author conducted. Rule/regulations are not perfect, so is your hazard analysis. If you want to compare the hazards you identified with those stated by rules/regulations, at least you should start with the same defined scope of analysis, the same type of accident (loss), the same hazards. I suggest that the authors rephrase these two paragraphs or can simply delete them.

14. Author's response to the preceding paragraph

The issues of imperfect analysis and comparison is correct. However, the paper doesn't attempt to provide comparisons with the rules or other analyses. As discussed in the earlier response, such comparison is indeed not possible because the input used for

drafting the rules is unavailable for public use.

However, this becomes irrelevant in the revised paper, which has slightly different scope.

6. The first paragraph in the conclusion part. I already stated my argument in the first comment about the objective. There is not enough evidence showing that risk assessments are done piecemeal and potentially lacking completeness. . .. It may sound better if you simply say that no such risk assessment has been conducted in the industry (only if experts from industry told you so). The value of academic work is not to conduct a specific risk assessment, sure you cannot find such from published papers. Conducting hazard analysis/risk assessment is an industrial practice which do not necessary provide enough academia value.

15. Author's response to the preceding paragraph

Agreed, the academic value from a hazard analysis is limited, if any. The response #7 alludes to the academic value of this work. The text throughout the paper will be updated accordingly.

7. The detailed intermediate results and final results from the STPA can be added in the appendix. For example, the control diagrams, identified unsafe control actions.

16. Author's response to the preceding paragraph

We have reduced the amount of graphics and the paper itself. Other results than those published cannot be shared due to PR constraints.

Please also note the supplement to this comment:
https://wes.copernicus.org/preprints/wes-2020-15/wes-2020-15-AC3-supplement.pdf

---

## Referee Report (RR1)

Review report of wes-2020-15 in title *"Revealing system variability of offshore service operations through systemic hazard analysis"*.

Overall comments:

The authors have addressed my previous comments and made big modifications to the paper.

Well, there still few issues that need to be concerned. Among them, there are two major issues.

1.  The objective of the paper is not well formulated.
    see following texts from the abstract.
    *"The objective of this paper is twofold. First, we perform a systemic hazard analysis by the STPA method for three phases of SOV operation: when transiting and maneuvering within a windfarm, interfacing with turbines, and launching or recovering daughter crafts. This gives us sets of scenarios containing potentially hazardous interactions between various system components. Such scenarios reflect the complexity and potential for necessary and unwanted variability in the system. Second, we use these results to compare the three operational phases in terms of a proposed systemic indicator—the system variability."*
    Both in the abstract and the introduction part, the authors only stated that "the objective of the paper is twofold" without specifying what are the objectives. The texts afterwards actually just a description of what you did (research process). The authors need to specify your research objective clearly.
    In my opinion, the objective in previous version is better at least it specified what you want to achieve.
    You also need to write your conclusion properly based on your objective.

2.  The author changed previous quantification to another systemic indicator-variability for quantification. The authors did not highlight why such a quantification (or measuring variability) is necessary or interesting for what. This should be the motivation of introducing quantification. Such motivation should be highlighted in the beginning of the paper.
    In my understanding, the general purpose of using STPA is to identify hazards and then to develop detailed control measures, so that people are not interested in quantification. If the authors want to introduce quantification or measure systemic variability of each phase, a reason should be given.
    In your conclusion part, you stated that "Knowledge where the system variability is highest, gives an opportunity to improve both performance (efficiency) and safety", Such can be a reason of why you want to introduce quantification of the systemic variability of different operational phases.

Minor comments

1.  The 3rd paragraph in the introduction part, "(section 2)" should be (section 3).

2. Section 7 is missing in the last paragraph of the introduction part where describe how the paper is organized.
3. The citation format in the second paragraph in Related work part does not seem right, see following texts:

   *"The reviewed literature focuses on collision (ship to ship, shop to turbine), reliability issues with technology (DP, gangway, and other systems) and human factors **(Presencia and Shafiee, 2018), (Dong et al., 2017), (Rollenhagen, 1997;Sklet, 2006), (Rokseth et al., 2017), and (SgurrEnergy, 2014)**."*

   The citation format needs to be check across the whole paper, I saw similar issues in many other places in the paper as well.
4. Some texts are too small in Figure 4 and Figure 7.
5. Figure 5 is a table or a figure? It seems to be a table.
6. The font in the paper is not the unified in the paper. Some figures have different fonts.
7. Language needs to be checked and refined again. Such as "Davit operator" or "David operator"? especially in Table 4, the verbs should be in plural or singular form?